# Assessment of Static Positioning Accuracy Using Low-Cost Smartphone GPS Devices for Geodetic Survey Points' Determination and Monitoring

**Marcin Uradziński** *[ID] and **Mieczysław Bakuła**

Faculty of Geoengineering, University of Warmia and Mazury, 10-719 Olsztyn, Poland; mieczyslaw.bakula@uwm.edu.pl
* Correspondence: marcin.uradzinski@uwm.edu.pl; Tel.: +48-89-523-48-72

**Abstract:** Recent developments enable to access raw Global Navigation Satellite System (GNSS) measurements of mobile phones. Initially, researchers using signals gathered by mobile phones for high accuracy surveying were not successful in ambiguity fixing. Nowadays, GNSS chips, which are built in the latest smartphones, deliver code and primarily carrier phase observations available for detailed analysis in post-processing applications. Therefore, we decided to check the performance of carrier phase ambiguity fixing and positioning accuracy results of the latest Huawei P30 pro smartphone equipped with a dual-frequency GNSS receiver. We collected 3 h of raw static data in separate sessions at a known point location. For two sessions, the mobile phone was mounted vertically and for the third one—horizontally. At the same time, a high-class geodetic receiver was used for L1 and L5 signal comparison purposes. The carrier phase measurements were processed using commercial post-processing software with reference to the closest base station observations located 4 km away. Additionally, 1 h sessions were divided into 10, 15, 20 and 30 min separate sub-sessions to check the accuracy of the surveying results in fast static mode. According to the post-processing results, we were able to fix all L1 ambiguities based on Global Positioning System (GPS)-only satellite constellation. In comparison to the fixed reference point position, all three 1 h static session results were at centimeters level of accuracy (1–4 cm). For fast static surveying mode, the best results were obtained for 20 and 30 min sessions, where average accuracy was also at centimeters level.

**Keywords:** smartphone; low-cost receivers; GNSS static positioning; GPS surveying

## 1. Introduction

The Global Navigation Satellite System (GNSS) chipsets built into the latest mobile phones, along with their miniaturization and reduced production costs, are nowadays a main aim of the electronic mass-market. Satellite positioning became a common practice in spreading of location-based applications dedicated for mobile devices, such as mobile phones, tablets or portable navigation systems. When the first Android Nougat 7 was introduced in May 2016, the GNSS observations could be processed by Android-dedicated applications and recorded into a file. Since then, the scientific community has been paying special attention to GNSS raw observations derived from smartphones. These observations include not only code pseudorange information but also carrier phase and Doppler GNSS measurements using Application Programming Interface (API). Such an interface defines interactions between multiple software intermediaries running on an Android operating system. Considering this solution, the stored GNSS observations are available now for analysis in post-processing.

Today, for navigational purposes, the typical accuracy obtained by a smartphone device is about a few meters, and under difficult conditions, may reach up to ten meters. It is caused by pseudoranges which are mainly used for low-precision real-time positioning [1]. To obtain precise positioning results, carrier-phase measurements are commonly used. Geodetic receivers with applied advanced positioning algorithms using multi-frequency GNSS observations may easily provide centimeter and sub-centimeter positioning results in real time or in post-processing mode.

Obtaining high positioning accuracies using GNSS technology can be achieved with the carrier phase observations and the capability to fix their ambiguities to the correct integer values. A short while ago, GNSS chipsets built into mobile phones provided mainly code and phase measurements only on the single frequency L1/E1. At the same time, the first post-processing algorithms were applied trying to obtain submeter positioning results with smartphones [2–4]. Developers started extracting raw measurements in a format which was best suited for post surveying processing and analysis. In the field of geodesy, there is a well-known interchange format for raw satellite navigation system data called RINEX (Receiver Independent Exchange Format). Considering current mobile applications such as RINEX ON or GEO++ RINEX Logger, there is an opportunity to collect GNSS data and conduct numerous research on post-processing positioning performances and signal analysis. The next significant technological breakthrough in smartphone positioning was in the year 2017, when the Broadcom company released the first dual-frequency GNSS chipset (BCM47755). Such a step was followed by other producers like Qualcomm or U-blox, who further presented chipsets (Snapdragon X24, Teseo, U-blox F9) using multi-constellation and multi-frequency technology [5,6]. The first smartphone equipped with a dual-frequency GNSS chipset was the Xiaomi Mi 8, introduced in June 2018. This device was the first equipped with the Broadcom BCM47755 GNSS dual-frequency chipset, designed by the Broadcom Limited company. The ability to use additional observations on a second frequency can increase signal availability and enables a higher level of positioning performances and accuracy by simplifying ambiguity fixing using wide-open techniques and corrections of most error sources, introduced mostly by ionospheric refraction. This strictly depends on the processing methodology and the current state of the atmosphere. The way to solve the carrier phase integer ambiguity enables introducing RTK (Real-Time Kinematic) or PPP (Precise Point Positioning) algorithms directly in mobile phones and the design of the additional L5/E5 signals help to recognize original signals from the ones reflected by near objects (multipath effect).

In the case of smartphones, GNSS observations suffer not only from high measurement noise, atmospheric refractions or multipath, but also from anomalies such as duty cycling and gradual accumulation of phase errors. Such a problem occurred with GNSS chipsets built in mobile phones, such as the Xiaomi Mi 8 [7–9]. The duty cycling is a process where the GNSS chipset of the smartphone works in a discontinuous way, which causes the hardware clock to be active only for a fraction of each second to support low power consumption and thus prevent battery discharge [10–13]. This action limited the use of smartphone observations for precise positioning techniques such as RTK and PPP. Fortunately, Google Nexus 9 was the first mobile device which enabled continuous recording of GNSS observations. First research results of the static data analysis showed that the horizontal and vertical RMS position errors were less than 0.8 and 1.4 m [4]. Other researchers [14,15] performed the studies using the same device. Position accuracy ranged from below 1 m to a few decimeters depending on experimental setups for positioning based on carrier-phase observations. Unfortunately, during signal analysis, various drifts for the code and phase observations appeared. In later Android versions, it was possible to turn off the duty cycling mode, making the observables applicable for phase-based positioning methods. Based on this, first single-baseline RTK positioning experiments considering two different kinds of multi-frequency and multi-constellation master stations took place [16]. This research work demonstrated that it is not possible to fix phase ambiguities and therefore to reach cm-accuracy in real-time, mostly due to the low quality of GNSS raw measurements collected by a mobile phone. For float-only solutions, researchers proved to obtain better precision with respect to the Global Positioning System (GPS)-only results. What they additionally realized,

when the GLONASS constellation is also added, the results are becoming worse. Additional works on recent GNSS smartphone positioning using single-frequency precise positioning in RTK or static PPP modes were presented in References [17,18]. Their investigation showed that the performance of current mobile phones is quite superior with respect to previous models of smart devices, but it is still not available to reach high coordinate accuracies.

The authors of this paper focused on assessment of carrier phase observations in precise smartphone positioning, and above all, in ambiguity fixing performances. Since November 2019, we have been doing research on various GNSS chipsets built into the mobile devices. Since then, we paid special attention to the latest dual-frequency GNSS Kirin 980 chipset, and in December 2019, we obtained a Huawei P30 Pro smartphone based on this chipset. In December 2019 and January 2020, we conducted several static measurement sessions in various configurations. Based on collected RINEX data, we analyzed the signals, the quality of carrier phase observations and finally single-base positioning results according to the nearest GNSS reference station.

## 2. Carrier Phase Observations for Static Measurements Provided by GNSS Chipsets

Obtaining high static positioning accuracies using GNSS technology strictly depends on carrier phase observations and the capability to fix their ambiguities to the correct integer values. Therefore, recently, scientists have been paying special attention to the analysis of such signals being tracked by smartphones. An example of raw measurements data collected by a Huawei P30 Pro smartphone (Union of Huawei Investment & Holding Co., Ltd., Shenzhen, China) is presented in Figure 1.

```
> 2020 01 22 08 00  0.0000384  0 26
G02 22925697.970      -66976.214        675.276       33.000
G03 24119241.297       94076.629      -3362.252       30.000   24119237.699   50408793.767   -2510.475    36.000
G04 21556455.876       70708.795      -2084.585       36.000
G05 24708637.166     -168006.751       3905.259       36.000
G06 22134843.966      603987.917      -1808.274       42.000   22134845.465   49976383.954   -1351.293    34.000
G07 22259767.783    -1088901.300       3016.162       34.000
G09 20231635.830       44531.941       -206.784       43.000
G16 23107344.918    63796500.344       1160.636       30.000
G23 21368845.756      507057.805      -1492.296       38.000
G26 23776544.341      248641.801       -834.344       42.000   23776543.741   49714144.959    -622.749    33.000
G29 24984080.481     -323748.564        806.904       41.000
G30 24691035.152    -1431120.291       4034.059       43.000   24691036.351   46908892.360    3014.874    27.000
R13 21636912.677  2183158326.800      -2618.043       30.000
R15 22316518.196   668008973.332       4636.240       45.000
R05 20017256.042   467466539.870       2415.095       42.000
R20 22351316.605   268041893.460      -2941.079       39.000
R21 22076500.757  2395270775.716        902.297       40.000
R03 23354599.347  2195522229.715      -3376.390       41.000
E08 22251034.230      318946.645       -939.008       33.000   22251030.332   64963702.784   -1214.096    23.000
E25 26958063.321      359157.562      -1091.169       30.000   26958055.527   65027061.841   -1426.759    23.000
R22 24635360.994   299230615.815       3906.437       27.000
E02 25060344.085    63765199.218       1295.820       35.000   25060342.886   47616869.697     967.753    37.000
E03 28428500.659    65407828.254      -2426.900       36.000   28428510.252   50166354.369   -1812.655    23.000
E07 23695537.721    63786199.097       1279.374       39.000   23695545.815   47632544.407     956.721    23.000
E26 24833227.015    65170081.670      -1821.123       34.000   24833232.411   49988839.291   -1359.885    30.000
E30 25349949.895    63065909.788       3321.301       22.000   25349955.291    -883948.183    2507.602    35.000
```

**Figure 1.** Single epoch for a Huawei P30 Pro smartphone collected in RINEX 3.03 (Receiver Independent Exchange Format 3.03).

As we can see from Figure 1, the smartphone recorded the following GNSS data:

- GPS L1, L5 carrier phase measurements transmitted at 1575.42 and 1176.45 MHz frequencies.
- GPS pseudoranges P(L1) and P(L5).
- GALILEO E1, E5a carrier phase measurements.
- GALILEO pseudoranges P(E1) and P(E5a).
- GLONASS L1 carrier phase measurements and pseudoranges.

It should be noted that the GPS L1 and L5 frequencies correspond with GALILEO E1 and E5a center frequencies. However, whereas all the satellites of GALILEO provide the L5 signal, in GPS, there are currently only 14 of them (where 12 are transmitted by satellites from the GPS IIF block and 2

from the GPS III block). In the case of double frequencies of L1–L5 GPS data in a relative positioning (static or RTK), we can write the following double-difference (DD) observation equations [19]:

$$\phi_{L1}(t) = \frac{\varrho(t)}{\lambda_{L1}} + N1 + \varepsilon_{\phi_{L1}}(t) \tag{1}$$

$$P(L1) = \varrho(t) + \varepsilon_{PL1}(t) \tag{2}$$

$$\phi_{L5}(t) = \frac{\varrho(t)}{\lambda_{L5}} + N5 + \varepsilon_{\phi_{L5}}(t) \tag{3}$$

$$P(L5) = \varrho(t) + \varepsilon_{PL5}(t) \tag{4}$$

where: $\phi_{L1}, \phi_{L5}$—observations of DD phase measurements for L1 and L5 frequencies (in cycles), P(L1) and P(L5)—observations of DD in code measurements (m), and $\varepsilon_{\phi_{L1}}, \varepsilon_{\phi_{L5}}, \varepsilon_{PL1}, \varepsilon_{PL5}$—errors of phase and code DD observations.

In some papers, mainly the signal-to-noise ratio has been analyzed, but we decided to focus on single epoch double-difference N1(P1) and N1(P5) float GPS-only solutions based on P(L1) and P(L5) codes, using Equations (1), (2) and (4), according to the formulas:

$$N1(P1) = \phi_{L1}(t) - \frac{P(L1)}{\lambda_{L1}} \tag{5}$$

$$N1(P5) = \phi_{L1}(t) - \frac{P(L5)}{\lambda_{L1}} \tag{6}$$

Time series of residuals of double-difference single epoch N1(P1) and N1(P5) GPS-only combinations for a geodetic receiver and smartphone is presented in Section 6.

Concerning the ratio of the power of the carrier phase signal to the power of noise, it should be noted that it is the most important factor depending on the class of receiver and type of antenna. There are limitations of embedded smartphone GNSS antennas addressed to the subject of low gain and low multipath elimination. Additionally, for precise positioning, mobile phone antennas require accurate phase center offset determination. Recent attempts focused on obtaining phase center information for the Huawei P30 were followed in Reference [20], where ambiguity fixing for a short baseline between the reference antenna and antenna being calibrated was required. After fixing ambiguities on L1 only, the authors showed that the antenna is located on the top of the smartphone (obtained with centimeters level of accuracy), which may be insufficient for precise static measurements. This is the main drawback to achieve high-accuracy positioning, especially because of problems with determining its exact location, which is usually not pointed out by the manufacturers.

In GNSS static measurements, we need to deal with additional factors concerning not only the type of equipment we use (receivers, antennas), but also with methods of planning and conducting the field surveys. The method of processing the GNSS measurements, such as the choice of correct methodology for processing GNSS baselines and managing a detailed analysis of their correct determination, also has an important influence on obtaining a reliable and accurate position. Only in the case of properly planned GNSS baselines and properly chosen reference station distance can we conduct a reliable accuracy analysis in post-processing static mode [21–23]. The baseline computation consists of a series of processing trails. Triple differencing is commonly used to determine and correct cycle slips in the datasets. A double differencing of the phase observables provides the baseline estimation. The latest post-processing algorithms involve highly advanced statistical methods for integer ambiguity determination. Combinations of the phase observations (narrow lane or wide lane) are often used as a step to confirm the final solution, according to baseline length.

Permanent GNSS reference stations, widely in use for baseline determination, allow us to model the distance-dependent errors (troposphere and ionosphere) and additionally, to verify the data transmitted in real time to the user, or downloaded for post-processing purposes. In 2008, the permanent reference

stations system (ASG-EUPOS) was launched in Poland offering Real-Time Kinematic (RTK) services and modules for Differential (DGNSS) surveys. This system also allows processing of the data from static measurements with a one-second measurement interval. The ASG-EUPOS reference-station system also provides virtual reference data in any location around the survey area [24]. Although, researchers found that virtual reference stations in post-processing mode do not increase the accuracy of GNSS survey at all [23]. This is related to the fact that the virtual reference stations located in reach of a few kilometers are very highly correlated to each other. When one is interested in centimeter/sub-centimeter accuracies, a rapid static survey technique will be much more reliable when conducted in relation to the local reference station [25]. Although, the virtual reference stations (VRS) in post-processing mode may still be the reference stations, in the case of rapid static measurements in which the session length is several minutes, the usefulness of a local permanent station is still recommended because the accuracy of the coordinates for the point being determined strictly depends on the accuracy of the reference station [26]. One should also mention that VRS observations rely on ionosphere and troposphere activity, which have a significant influence in rapid static surveys [27]. Therefore, for results comparison and reliability of the accuracy for the coordinates determined, the authors decided that all measurements using rapid static techniques will be much more reliable with reference to the local ASG EUPOS permanent station. In static relative positioning, both using a real reference station or a virtual one, the efficiency of determining precise baselines will be primarily affected by the accuracy of the phase measurements of the mobile receiver (in this case, smartphone) due to the quality of the phone's phase observations.

## 3. Use of GNSS Carrier Phase Observations in Various Fields of Science

While considering carrier phase GNSS observations, one should also mention their practical applications. Next to geodetic and topographic surveys, the static method is widely used to calculate high accuracy three-dimensional coordinates in traverse stations for objects such as roads or railways, accurately determining the azimuth for establishing the network's orientation or setting points for monitoring deformations of objects. A mobile GNSS-based positioning system provides an excellent tool to quickly, accurately and reliably position points of details or features which may need to be precisely mapped. Detail surveys typically require an accuracy between 1 and 10 cm. There are also some applications, like utility mapping, which require decimeter accuracies.

The usefulness of geodetic techniques, in particular GNSS technology, for monitoring different kinds of deformations is also a common practice. This is usually performed by setting a network of geodetic satellite receivers, obtaining accuracies in the order of sub-centimeters. The use of a low-cost u-blox GPS device may be an example, showing that good results for such applications can be achieved [28]. Other researchers presented wireless monitoring systems, consisting of GNSS receivers and accelerometers, for long-term monitoring installations, landslide monitoring or efficient and effective reconstruction of the railway and tram tracks' geometry [29–32]. In the case of constructions of super-high buildings, satellite-based techniques may be very important to help control them vertically and also analyze their horizontal motions [33].

More recent research demonstrates the advantage of the integration of mass-market smartphone GNSS signals with other precise sensors for geoscience applications, where accelerometers and gyroscopes are additionally used for earthquake detection or ionospheric/tropospheric disturbances [34,35].

GNSS technology also puts forward the foundations for atmospheric precipitable water vapor analysis. Compared to the conventional methods of atmospheric water vapor detection (such as radio sounding, microwave radiometer, satellite remote sensing, etc.), GNSS-based atmospheric water vapor detection has high precision and high space–time resolution and became a powerful tool which is widely used nowadays. Since the water vapor may be easily derived from inverting tropospheric delays, which is estimated in GNSS post-processing, the outcomes may be incorporated into the analysis of the rainfall process to support the weather forecast [36,37]. Additionally, scientists have been trying to provide novel methods to measure the impact of ionosphere on GNSS measurements. Despite being

one of the major sources of error affecting high-accuracy GNSS applications, ionospheric scintillations are an important tool to study the behavior of this layer of the atmosphere [38].

Finally, low-cost GNSS receivers have been widely used for precise navigation purposes. Delivering raw dual frequency data, receivers allow us a more sophisticated processing, such as the double-difference approach, and therefore, a more accurate positioning. However, high accuracies can be generally obtained only with an open sky visibility, without any obstructions. The problem appears while positioning in urban environments or being indoors, even using low-cost GNSS receivers equipped with highly sensitive antennas. In such situations, satellite receivers should be integrated with additional vision or microelectromechanical (MEMS) sensors [39,40]. With the development of global satellite navigation systems, kinematic Precise Point Positioning (PPP) is one of the recent techniques to make positioning efficient and cost effective by reducing equipment costs for surveying purposes. It does not require any additional data from a reference station and can provide a solution with a centimeter to decimeter level of position accuracy both in static and kinematic modes [41,42]. Therefore, PPP may be widely used in many fields, such as precise navigation or surveying.

## 4. Equipment and Surveying Methodology

The static observation sessions with use of the Huawei P30 Pro smartphone were conducted on three different days at the same reference point located in the open field area at the University of Warmia and Mazury in Olsztyn. All surveying session data was collected in the end of January 2020 on three separate days, starting at the same local time. Although we presented the results of static GPS positioning based on three 1 h measurement sessions, we actually performed longer additional tests before, also focusing on short baseline solutions, where the accuracy results were very similar to the presented ones. Before we started the final tests, we tried out different mobile RINEX loggers to check the quality of data. Nowadays, the most popular available applications are GEO++ RINEX Logger (Version 2.1.6, Geo++ GmbH, Garbsen, Germany, 2020) and RINEX ON (Nottingham Scientific Ltd, Nottingham, United Kingdom, 2020). Since data collected by two of the same Huawei P30 pro devices using GEO++ RINEX logger was poor quality after data and signal analysis using Topcon Tools commercial software (Version 7.5, Topcon Corporation, Tokyo, Japan, 2009) (Figure 2), we decided to use the RINEX ON application for further GNSS raw data measurements. The Huawei P30 Pro smartphone has a possibility to collect data from GPS, GLONASS, BEIDOU and GALILEO positioning systems. However, due to the Topcon Tools v.7.5 software used, it was possible to process GPS plus GLONASS, and GPS-only observations. GPS plus GLONASS calculations were made, but no better results were obtained than with GPS-only. Therefore, we decided to present GPS-only positioning results. The latest commercial software packages enable static and kinematic data processing with the integration of all GNSS systems. Therefore, this topic will be the subject of further research by the authors.

For comparison purposes, we firstly occupied a fixed control point in the research area with a GNSS Javad Alpha geodetic receiver (3 h session) to obtain accurate reference coordinates. Baseline post-processing was performed using Topcon Tools v.7.5 with reference to the local OPNT (Olsztynski Park Naukowo-Technologiczny) permanent GNSS station (ASG EUPOS), located 4 km away.

The first surveying to assess the usefulness of the Huawei P30 Pro was conducted on 15 January 2020. Data was collected over 60 min and the mobile phone was mounted in vertical position and centered over a reference point (Figure 3).

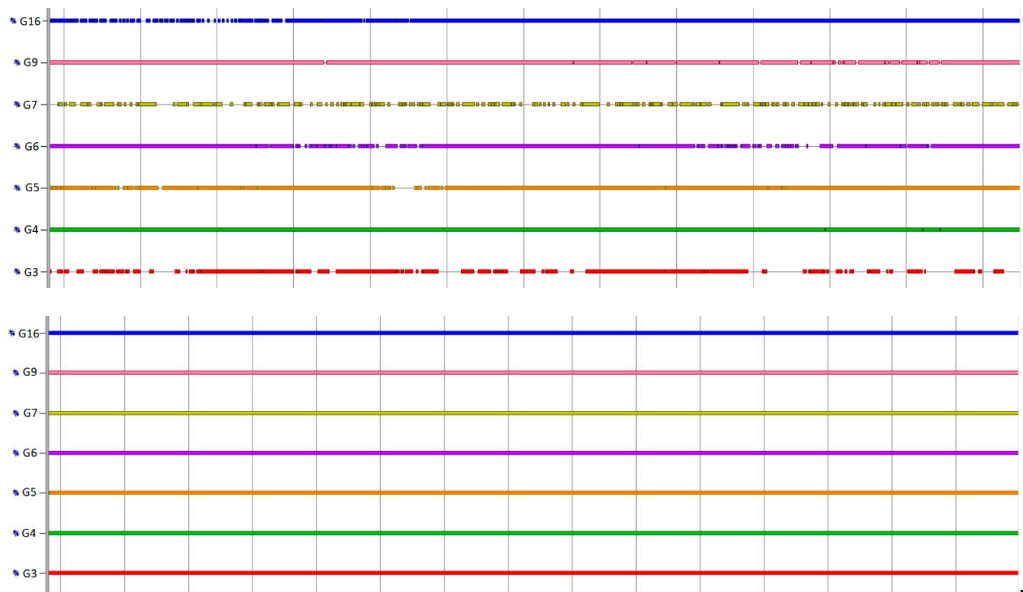

**Figure 2.** Comparison of data quality collected by GEO++ RINEX Logger (upper) and RINEX ON (bottom).

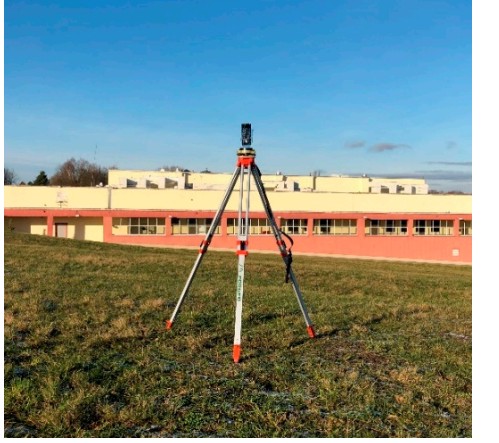 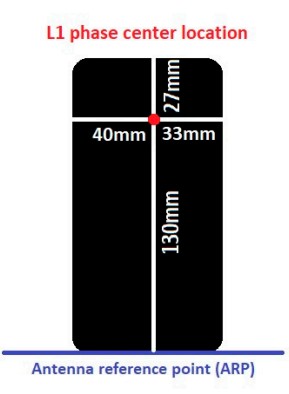

**Figure 3.** Test area and the initial research on Huawei P30 Pro static positioning accuracy. Huawei P30 Pro L1 phase center location.

To obtain centimeter level of accuracy using carrier-phase observations, there is a need for precise information on average phase center position of internal antenna and possible phase center variations. Since the specification of the used smartphone does not include any information on GNSS antenna location, we did some research on the internet and contacted technical services and learnt that the antenna phase center is located at the top of device, in the line of the upper camera. This information was confirmed later in our research on phase center, which was determined by us very precisely (millimeter level of accuracy). We conducted three separate experiments (2 h sessions) for this purpose, where the mobile phone was placed on the base made of an aluminum beam with centrally positioned mandrel that allows for mounting it on the levelling head, which is centered over the reference point. At the same time, when the central device is positioned above the point being determined, the other external receivers may be set at both the edge sides at a constant distance of 0.50 m from the central one. The central Huawei P30 pro smartphone was fixed in three various positions on the aluminum base. Firstly, mounted vertically, then, lying along the beam (parallel) and for the last option—perpendicularly to it. The base was equipped with a professional sensitive compass allowing the positioning of the base along the north–south line. At the same time, for all phase-center determination measurements, two additional GNSS Javad Alpha receivers were mounted at the north-end and south-end locations

of the aluminum beam. Results of phase center determination using L1 fixed baseline solutions are additionally presented in Figure 3. GNSS data in a smartphone was collected using RINEX ON Android application in RINEX version 3.03. What we realized while collecting data using this application is that during this process, the smartphone's screensaver should be turned off. Otherwise, data being recorded will be interrupted or of poor quality. Since the latest RINEX version acceptable in Topcon Tools v.7.5 software is 3.00, we had to make a conversion of RINEX v3.03 to v3.00 files using RTKLib v2.4.3 software (open source program package, 2020). The computation of baseline distances between Javad receivers and the Huawei P30 Pro for all three cases were obtained with millimeter level of precision. After calculating all the offsets, we received precise information as to where the phase center of the internal antenna is located.

Two further static tests also lasted for 60 min each. The first one was conducted on 22 January, and the Huawei P30 Pro smartphone was also mounted in vertical position. Unlike in the initial test, the mobile phone was placed on an aluminum beam, and centered above the control point with offset of 0.02 m (Figure 4). Additionally, for signal quality analysis, a Javad Alpha GNSS geodetic receiver (Javad GNSS, San Jose, CA, USA) was placed 0.52 m away from a smartphone, in the north direction.

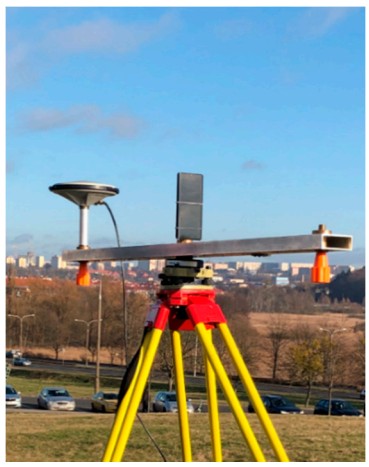

**Figure 4.** Second test setup. Huawei P30 Pro mounted in vertical position.

The last test took place on 23 January, starting at the same local time. In this case, the smartphone was mounted in a horizontal position, centered above the reference point, with offset of 0.155 m according to the determined phase center of the GNSS antenna (Figure 5). As in the previous experiment, the Javad receiver was mounted at the same mandrel.

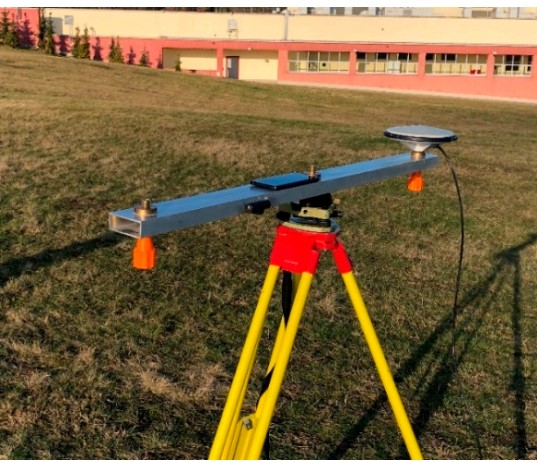

**Figure 5.** Third test setup. Huawei P30 Pro mounted in a horizontal position.

## 5. Smartphone Positioning Results

Considering the first positioning test on 15 January, a Huawei P30 Pro was mounted in vertical position on a tripod. Static coordinate results were computed based on carrier phase observations obtained from the OPNT permanent station (4 km away) and compared to the reference position. We subdivided the observation data into sessions of shorter duration of 10, 15, 20, 30 and ending on 60 min. For all time intervals, the best results were obtained with GPS-only L1 carrier phase observations, where all the ambiguities were fixed. The computation of baseline coordinates in all cases was provided by Topcon Tools software. During the 1 h session, the number of observed GPS satellites did not drop below 6, with an average number of 8 satellites. Figures 6–10 and Tables 1–5 present results of the first initial test, after a comparison to the reference position.

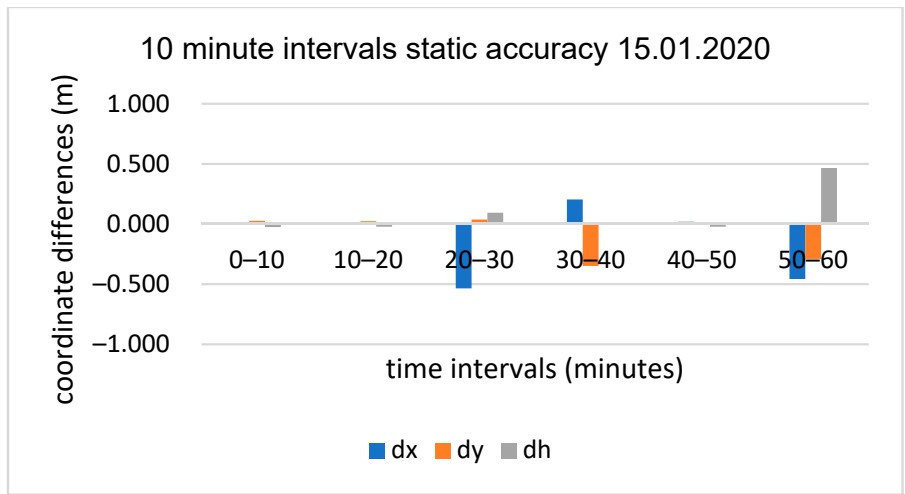

**Figure 6.** 10 min sub-session positioning results for initial test, baseline OPNT–P30Pro (4 km). OPNT: Olsztynski Park Naukowo-Technologiczny; dx, dy, dh: coordinate differences between the true position and smartphone static results in north (dx), east (dy) and height (dh) components, respectively.

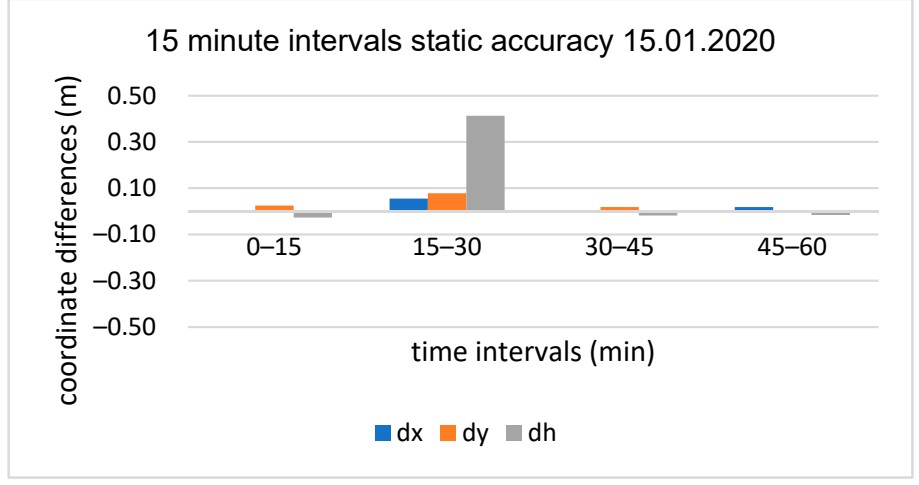

**Figure 7.** 15 min sub-session positioning results for initial test, baseline OPNT–P30Pro (4 km).

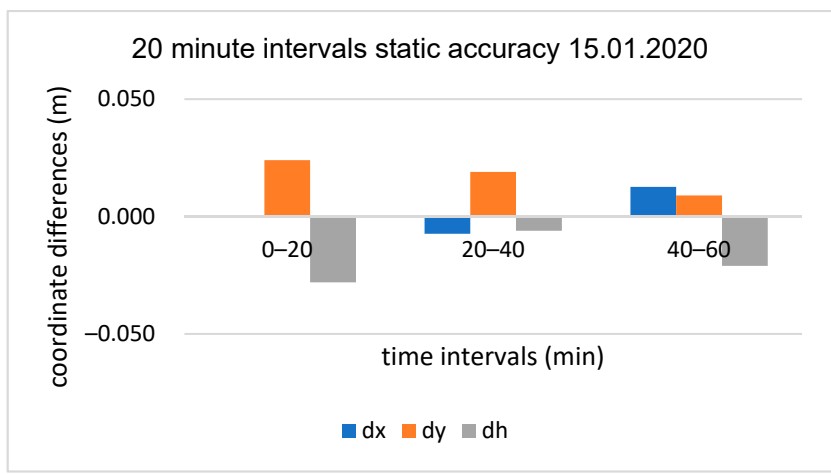

**Figure 8.** 20 min sub-session positioning results for initial test, baseline OPNT–P30Pro (4 km).

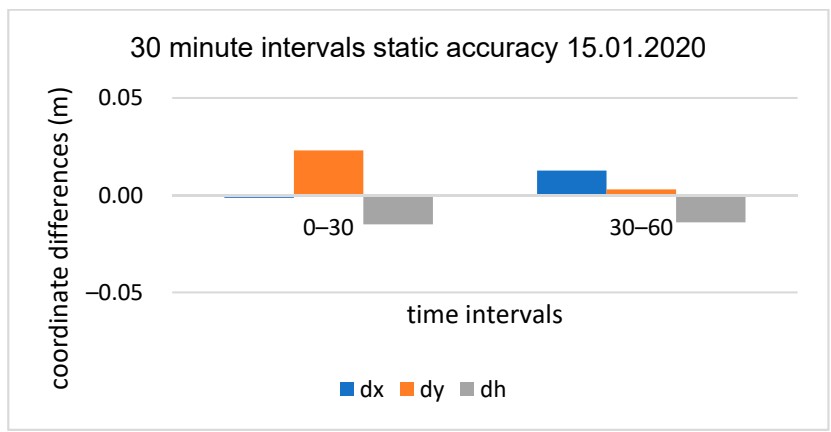

**Figure 9.** 30 min sub-session positioning results for initial test, baseline OPNT–P30Pro (4 km).

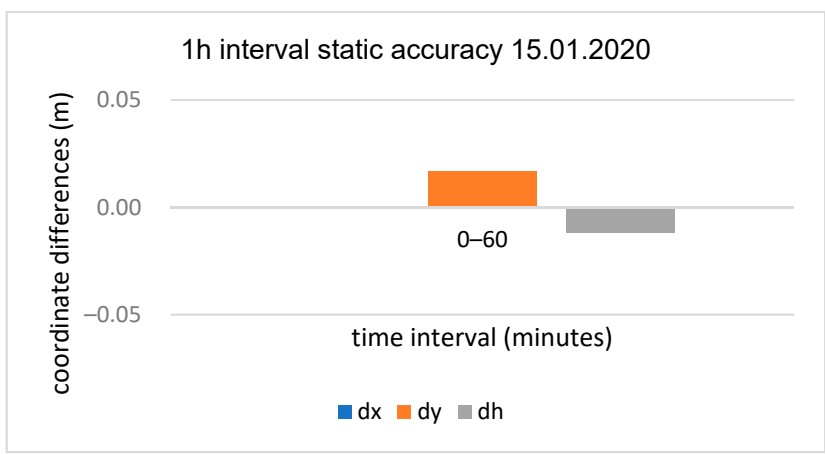

**Figure 10.** 60 min session positioning results for initial test, baseline OPNT–P30Pro (4 km).

**Table 1.** 10 min sub-session positioning results summary, baseline OPNT–P30Pro (4 km).

| Minutes | dx (m) | dy (m) | dh (m) |
|---------|--------|--------|--------|
| 0–10    | 0.006  | 0.026  | −0.025 |
| 10–20   | −0.004 | 0.024  | −0.022 |
| 20–30   | −0.534 | 0.038  | 0.094  |
| 30–40   | 0.205  | −0.348 | −0.006 |
| 40–50   | 0.020  | 0.005  | −0.022 |
| 50–60   | −0.457 | −0.290 | 0.466  |
| min     | −0.534 | −0.348 | −0.025 |
| max     | 0.205  | 0.038  | 0.466  |
| SD      | 0.296  | 0.178  | 0.194  |

SD: standard deviation of coordinates in north (x), east (y) and height (h) components, respectively; dx, dy, dh: coordinate differences between the true position and smartphone static results in north (dx), east (dy) and height (dh) components, respectively.

**Table 2.** 15 min sub-session positioning results summary, baseline OPNT–P30Pro (4 km).

| Minutes | dx (m) | dy (m) | dh (m) |
|---------|--------|--------|--------|
| 0–15    | 0.005  | 0.025  | −0.027 |
| 15–30   | 0.055  | 0.078  | 0.413  |
| 30–45   | 0.002  | 0.018  | −0.018 |
| 45–60   | 0.019  | 0.002  | −0.016 |
| min     | 0.002  | 0.002  | −0.027 |
| max     | 0.055  | 0.078  | 0.413  |
| SD      | 0.025  | 0.033  | 0.217  |

**Table 3.** 20 min sub-session positioning results summary, baseline OPNT–P30Pro (4 km).

| Minutes | dx (m) | dy (m) | dh (m) |
|---------|--------|--------|--------|
| 0–20    | 0.000  | 0.024  | −0.028 |
| 20–40   | −0.007 | 0.019  | −0.006 |
| 40–60   | 0.013  | 0.009  | −0.021 |
| min     | −0.007 | 0.009  | −0.028 |
| max     | 0.013  | 0.024  | −0.006 |
| SD      | 0.010  | 0.008  | 0.011  |

**Table 4.** 30 min sub-session positioning results summary, baseline OPNT–P30Pro (4 km).

| Minutes | dx (m) | dy (m) | dh (m) |
|---------|--------|--------|--------|
| 0–30    | −0.001 | 0.023  | −0.015 |
| 30–60   | 0.013  | 0.003  | −0.014 |
| min     | −0.001 | 0.003  | −0.015 |
| max     | 0.013  | 0.023  | −0.014 |
| SD      | 0.010  | 0.014  | 0.001  |

**Table 5.** 60 min session positioning results summary, baseline OPNT–P30Pro (4 km).

| Minutes | dx (m) | dy (m) | dh (m) |
|---------|--------|--------|--------|
| 0–60    | 0.0007 | 0.017  | −0.012 |

In the second positioning test on 22 January, a Huawei P30 Pro was mounted, also in a vertical position, but on an aluminum beam. The complete surveying session lasted 60 min. As mentioned before, for signal quality comparisons, a Javad Alpha geodetic receiver was mounted in the north-end location, 0.50 m away from a mobile phone. Static coordinate results were also computed based on carrier phase observations obtained from the OPNT permanent station and compared to the reference position. Again, we subdivided the observation data into sessions of shorter duration. Based on previous experience, GPS-only L1 carrier phase observations where used to process collected data in Topcon Tools post-processing software. The same as in the previous experiment, all the ambiguities were fixed. During this session, the number of observed GPS satellites did not drop below 5, and the average number was the same as before (8 satellites). Results of this experiment are presented in the Figures 11–15 and Tables 6–10 below.

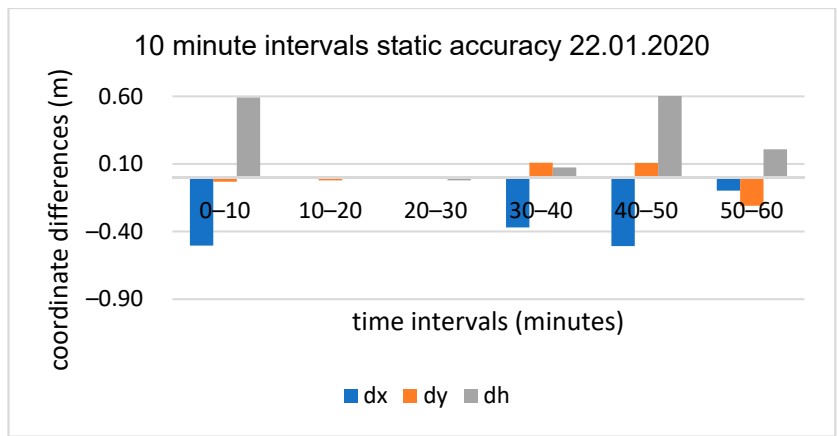

**Figure 11.** 10 min sub-session positioning results for the second test, baseline OPNT–P30Pro (4 km).

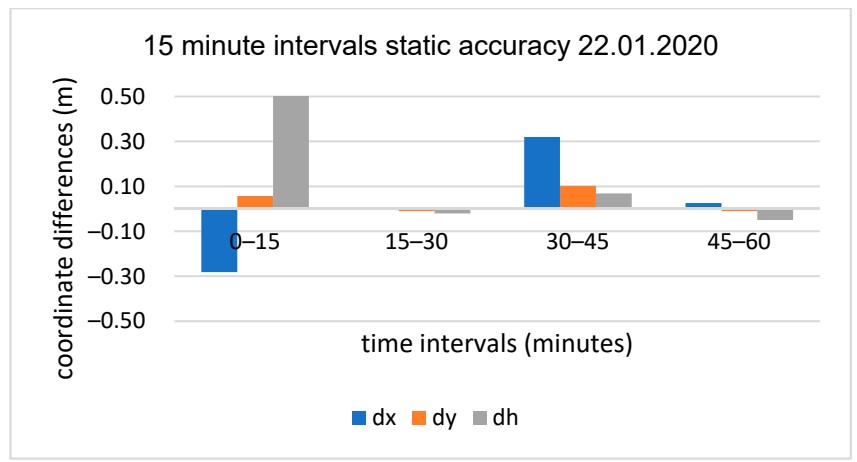

**Figure 12.** 15 min sub-session positioning results for the second test, baseline OPNT–P30Pro (4km).

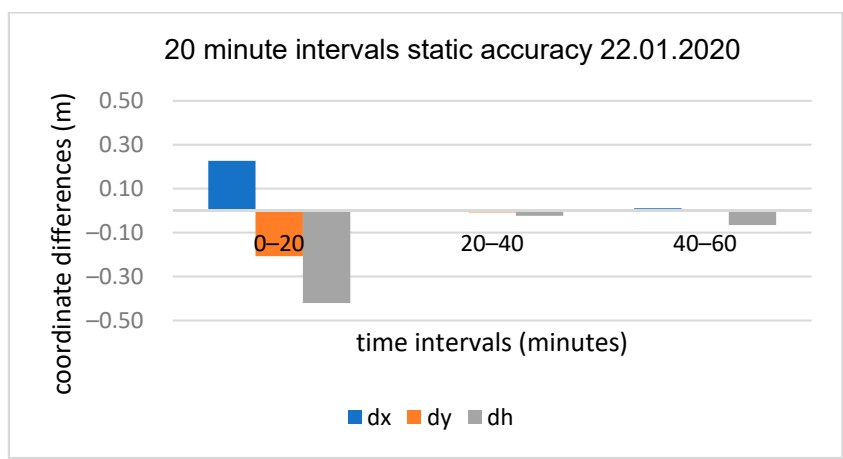

**Figure 13.** 20 min sub-session positioning results for the second test, baseline OPNT–P30Pro (4 km).

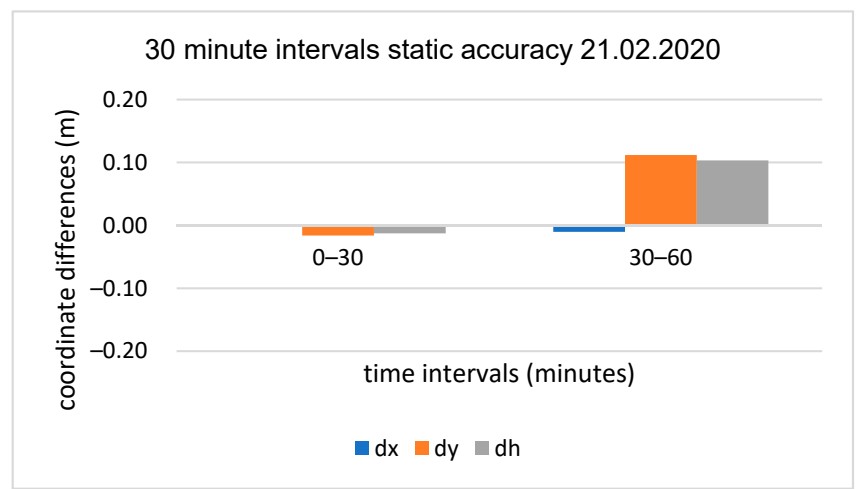

**Figure 14.** 30 min sub-session positioning results for the second test, baseline OPNT–P30Pro (4 km).

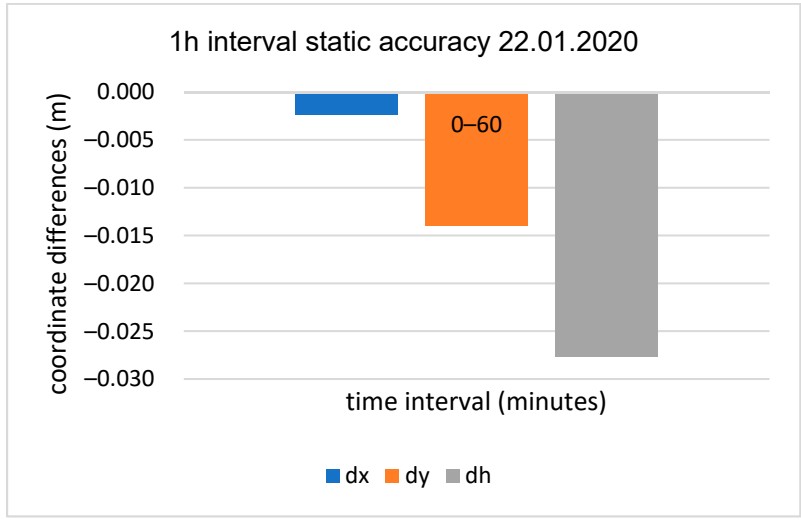

**Figure 15.** 60 min session positioning results for the second test, baseline OPNT–P30Pro (4 km).

**Table 6.** 10 min sub-session positioning results summary, baseline OPNT–P30Pro (4 km).

| Minutes | dx (m) | dy (m) | dh (m) |
|---|---|---|---|
| 0–10 | −0.504 | −0.032 | 0.591 |
| 10–20 | 0.002 | −0.022 | −0.003 |
| 20–30 | 0.001 | −0.009 | −0.022 |
| 30–40 | −0.369 | 0.108 | 0.074 |
| 40–50 | −0.507 | 0.107 | 0.767 |
| 50–60 | −0.097 | −0.209 | 0.208 |
| min | −0.507 | −0.209 | −0.022 |
| max | 0.002 | 0.108 | 0.767 |
| SD | 0.243 | 0.116 | 0.332 |

**Table 7.** 15 min session positioning results summary, baseline OPNT–P30Pro (4 km).

| Minutes | dx (m) | dy (m) | dh (m) |
|---|---|---|---|
| 0–15 | −0.281 | 0.057 | 0.545 |
| 15–30 | 0.002 | −0.010 | −0.021 |
| 30–45 | 0.320 | 0.102 | 0.068 |
| 45–60 | 0.026 | −0.010 | −0.050 |
| min | −0.281 | −0.010 | −0.050 |
| max | 0.320 | 0.102 | 0.545 |
| SD | 0.246 | 0.055 | 0.278 |

**Table 8.** 20 min sub-session positioning results summary, baseline OPNT–P30Pro (4 km).

| Minutes | dx (m) | dy (m) | dh (m) |
|---|---|---|---|
| 0–20 | 0.227 | −0.207 | −0.421 |
| 20–40 | 0.006 | −0.008 | −0.024 |
| 40–60 | 0.011 | −0.003 | −0.066 |
| min | 0.006 | −0.207 | −0.421 |
| max | 0.227 | −0.003 | −0.024 |
| SD | 0.126 | 0.116 | 0.218 |

**Table 9.** 30 min sub-session positioning results summary, baseline OPNT–P30Pro (4 km).

| Minutes | dx (m) | dy (m) | dh (m) |
|---|---|---|---|
| 0–30 | −0.002 | −0.016 | −0.013 |
| 30–60 | −0.010 | 0.112 | 0.103 |
| min | −0.010 | −0.016 | −0.013 |
| max | −0.002 | 0.112 | 0.103 |
| SD | 0.006 | 0.091 | 0.082 |

**Table 10.** 60 min session positioning results summary, baseline OPNT–P30Pro (4 km).

| Minutes | dx (m) | dy (m) | dh (m) |
|---|---|---|---|
| 0–60 | −0.002 | −0.014 | −0.028 |

Using RTKLib GNSS software, we prepared a skyplot covering the GPS satellites tracked by the Huawei P30 Pro (Figure 16).

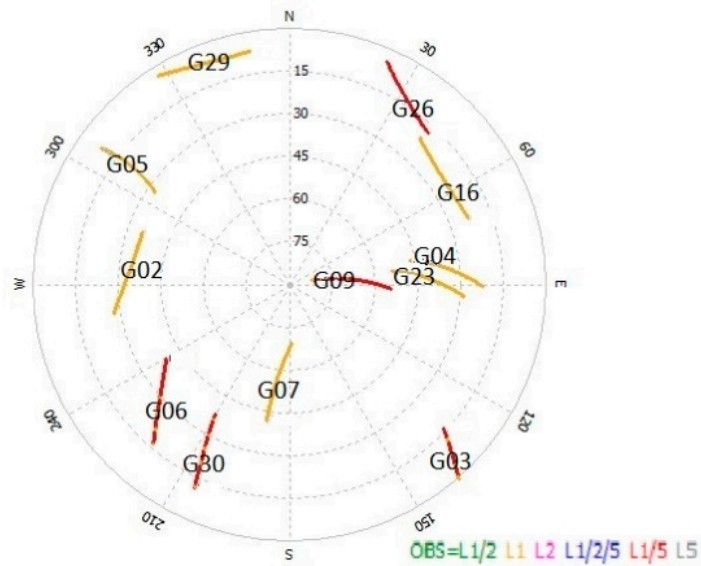

**Figure 16.** Huawei P30 Pro skyplot of Global Positioning System (GPS)-only.

In the last positioning test, which took place on 23 January, the Huawei P30 Pro was placed horizontally on an aluminum beam. Session time was also 60 min and started at the same time as the day before. Similarly, we mounted the Javad Alpha geodetic receiver at the same place. Static coordinate results were also computed based on carrier phase observations obtained from the OPNT permanent station and compared to the reference position. Again, we subdivided the observation data into sessions of shorter duration. Based on previous experience, GPS-only L1 carrier phase observations where used to process collected data in Topcon Tools post-processing software. Unfortunately, in the case of the 10 min sub-session, all the ambiguities were float. The same problem occurred in 15 min and 20 min sessions. Only for 30 and 60 min sessions were ambiguities fixed. During this 60 min surveying session, the number of observed GPS satellites did not drop below 6, and the average number was the same as before (8 satellites). Results of this experiment are presented in Figures 17–21 and Tables 11–15 below.

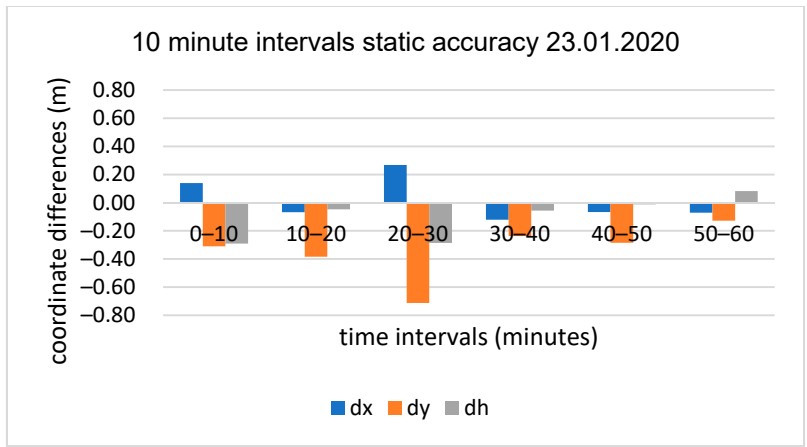

**Figure 17.** 10 min sub-session positioning results for the third test, baseline OPNT–P30Pro (4 km).

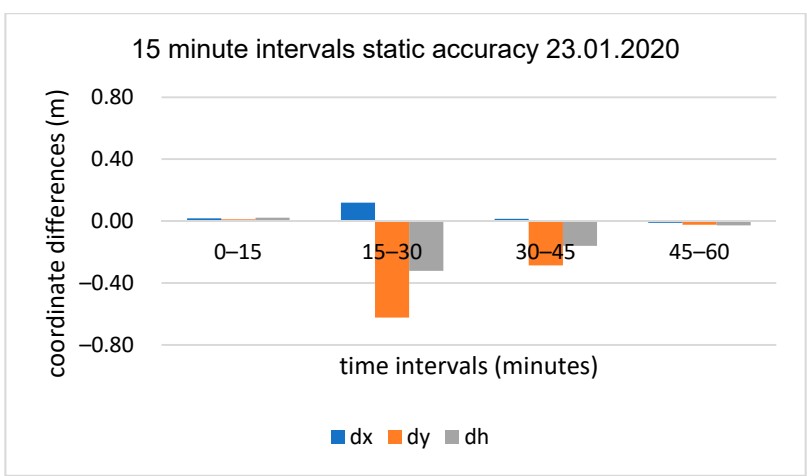

**Figure 18.** 15 min sub-session positioning results for the third test, baseline OPNT–P30Pro (4 km).

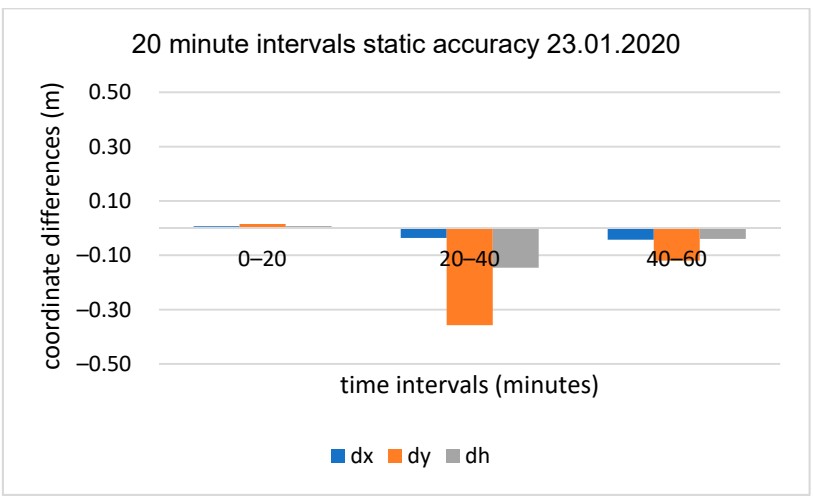

**Figure 19.** 20 min sub-session positioning results for the third test, baseline OPNT–P30Pro (4 km).

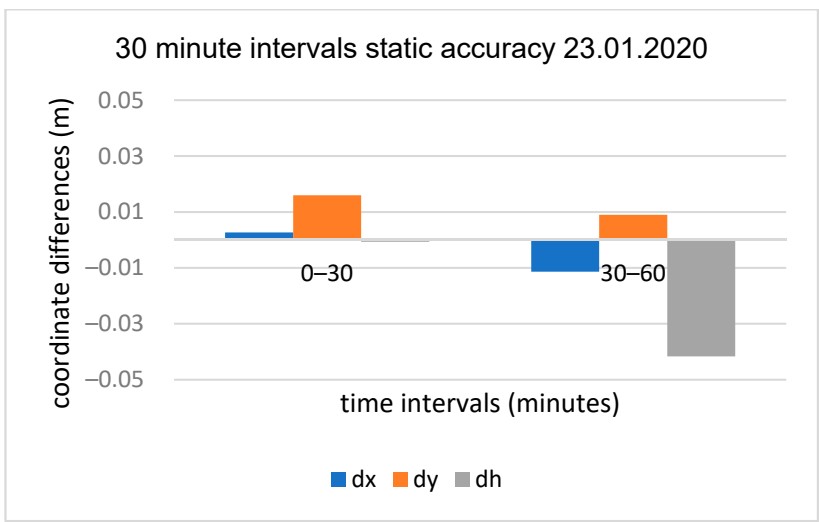

**Figure 20.** 30 min sub-session positioning results for the third test, baseline OPNT–P30Pro (4 km).

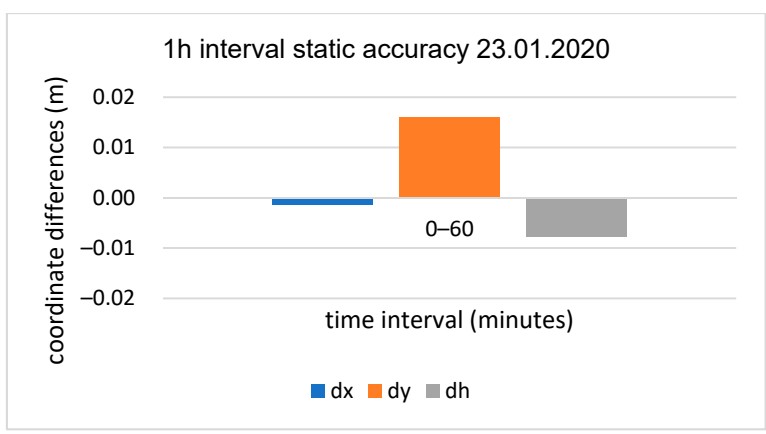

**Figure 21.** 60 min session positioning results for the third test, baseline OPNT–P30Pro (4 km).

**Table 11.** 10 min sub-session positioning results summary, baseline OPNT–P30Pro (4 km).

| Minutes | dx (m) | dy (m) | dh (m) |
|---------|--------|--------|--------|
| 0–10 | 0.139 | −0.310 | −0.291 |
| 10–20 | −0.068 | −0.384 | −0.048 |
| 20–30 | 0.268 | −0.713 | −0.287 |
| 30–40 | −0.121 | −0.237 | −0.057 |
| 40–50 | −0.066 | −0.285 | −0.012 |
| 50–60 | −0.071 | −0.128 | 0.082 |
| min | −0.121 | −0.713 | −0.291 |
| max | 0.268 | −0.128 | 0.082 |
| SD | 0.154 | 0.200 | 0.153 |

**Table 12.** 15 min sub-session positioning results summary, baseline OPNT–P30Pro (4 km).

| Minutes | dx (m) | dy (m) | dh (m) |
|---------|--------|--------|--------|
| 0–15 | 0.018 | 0.011 | 0.021 |
| 15–30 | 0.119 | −0.624 | −0.322 |
| 30–45 | 0.015 | −0.286 | −0.161 |
| 45–60 | −0.011 | −0.024 | −0.028 |
| min | −0.011 | −0.624 | −0.322 |
| max | 0.119 | 0.011 | 0.021 |
| SD | 0.057 | 0.294 | 0.154 |

**Table 13.** 20 min sub-session positioning results summary, baseline OPNT–P30Pro (4 km).

| Minutes | dx (m) | dy (m) | dh (m) |
|---------|--------|--------|--------|
| 0–20 | 0.007 | 0.015 | 0.007 |
| 20–40 | −0.036 | −0.357 | −0.146 |
| 40–60 | −0.042 | −0.120 | −0.040 |
| min | −0.042 | −0.357 | −0.146 |
| max | 0.007 | 0.015 | 0.007 |
| SD | 0.027 | 0.188 | 0.078 |

**Table 14.** 30 min sub-session positioning results summary, baseline OPNT–P30Pro (4 km).

| Minutes | dx (m) | dy (m) | dh (m) |
|---------|--------|--------|--------|
| 0–30    | 0.003  | 0.016  | −0.001 |
| 30–60   | −0.011 | 0.009  | −0.042 |
| min     | −0.011 | 0.009  | −0.042 |
| max     | 0.003  | 0.016  | −0.001 |
| SD      | 0.010  | 0.005  | 0.029  |

**Table 15.** 60 min session positioning results summary, baseline OPNT–P30Pro (4 km).

| Minutes | dx (m) | dy (m) | dh (m) |
|---------|--------|--------|--------|
| 0–60    | −0.001 | 0.016  | −0.008 |

The Huawei P30 Pro GPS-only skyplot for the third experiment is presented in Figure 22.

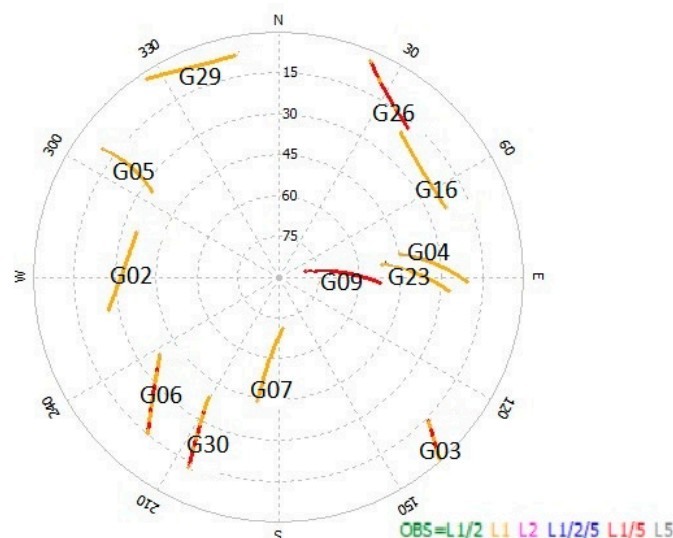

**Figure 22.** Huawei P30 Pro skyplot of GPS-only (third experiment).

## 6. Discussion and Summary

In the conducted GNSS measurements, a mobile phone was used with the option of data recording on two frequencies, L1/E1 and L5/E5a, and a possibility of registering GNSS observations from GPS and GALILEO systems, and L1 observations for the GLONASS system. This created a possibility to track over 25 satellites simultaneously during the test measurements, of which more than 10 satellites had observations on L1/E1–L5/E5a. Unfortunately, GALILEO observations were not recorded at the reference station. Therefore, only L1 frequency observations for the GPS system were used for comparative calculations in the described tests. In addition, GPS plus GLONASS calculations were made, but no better results were obtained than with GPS-only. Generally, some centimeters accuracy was obtained at the level of the professional geodetic receiver for 30 min sessions, and in some cases, for 10 min sessions. This means that, undoubtedly, a new era has arrived for GNSS systems, where every mobile phone user will have the opportunity to obtain a position with some centimeters accuracy. What is more, mobile phone manufacturers even overtook the owners of reference station systems by providing L1–L5 observations from the GALILEO system. This creates much greater positioning options, both for autonomous positioning, as well as for static measurements or RTK techniques.

Figure 23 shows time series of residual values of float DD ambiguities (N1), including code observations P(L1) and P(L5), for the baselines of OPNT-JAVAD and OPNT-Huawei P30. The standard

deviation (SD) for the P(L1) code measurements was 2.6 m for the Huawei P30 Pro and 0.8 m for the JAVAD receiver, while for the P(L5) code, it was 0.8 and 0.5 m, respectively. It can be seen that if the L1 code measurements are inferior to the Javad GNSS receiver, the P(L5) code measurements are at a similar level of accuracy, only slightly differing from the professional GNSS receiver used. This creates great opportunities in the future for the use of smartphones based on the P(L5) code. Based on this information, this code should and probably will be mainly used for GNSS positioning, enabling even decimeter accuracy of autonomous positioning. In addition, such high accuracy of the P(L5) code definitely increases the possibilities for determining ambiguities for single observation epochs of phase observations L1/E1–L5/E5a of GPS and GALILEO systems. However, it should be noted that the phase observations L1–L5 are currently much worse for the tested Huawei P30 pro compared to those of a professional GNSS receiver. The ambiguity values for N1–N5 solutions from single measurement epochs based on the use of only one pair of G06–G26 satellites are shown in Figure 24. For the OPNT-Javad baseline, we have all float ambiguities arranged precisely along the x-axis, while for the OPNT-Huawei P30 Pro baseline, it can be seen that some phase observations have errors that can be understood as gross errors. However, many of the ambiguities are located along the line, and these are good observations that allow to achieve some centimeters accuracy.

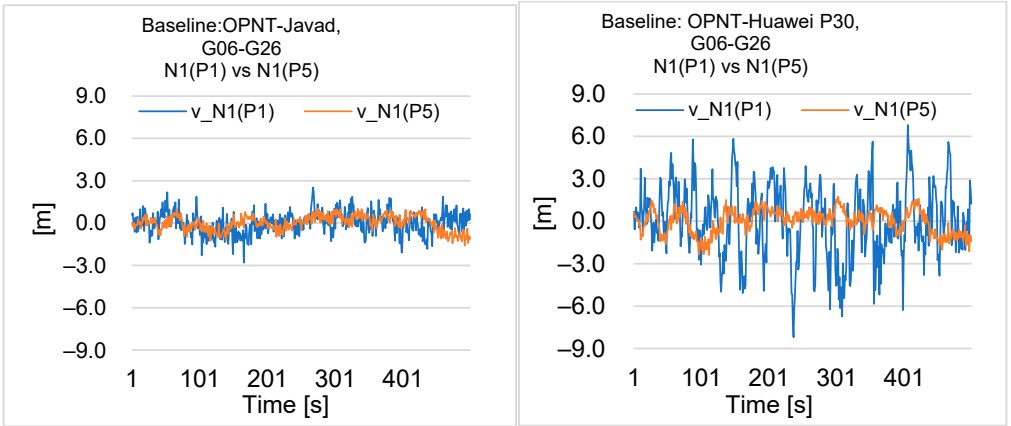

**Figure 23.** Time series of double-difference residuals for float N1(P1) and N1(P5) solutions from single measurement epochs based on the use of only one pair of G06–G26 satellites (500 epochs).

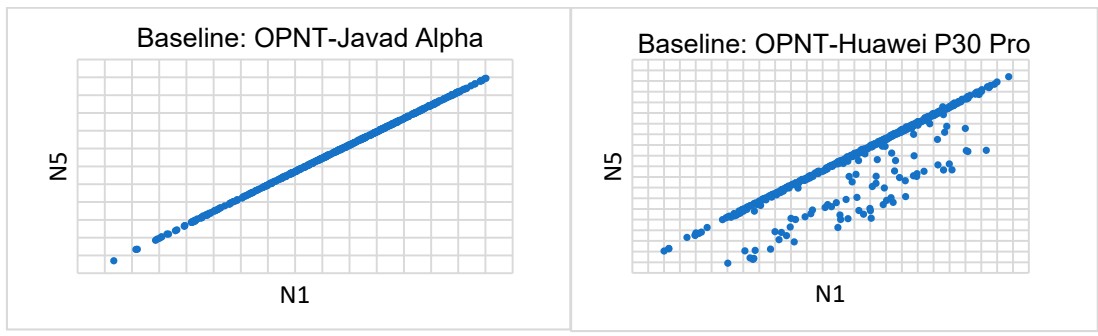

**Figure 24.** Time series of correlated double-difference float single epoch N1–N5 solutions for a pair of G06–G26 satellites (500 epochs).

The unfixed ambiguities with observation periods of just a few minutes are therefore caused by the low quality of the carrier phase observations at the moment of observation. Even when we consider perfect environmental conditions, the number of data gaps, noise level and multipath effect are moderately larger as compared to a professional GNSS receiver. This could have a direct effect on the quality of float solutions in some cases, and afterwards, on the ability to fix carrier phase ambiguities to their true values.

**Author Contributions:** Conceptualization, M.U. and M.B.; methodology, M.U.; software, M.B.; validation, M.U., M.B.; formal analysis, M.U.; investigation, M.B.; resources, M.U.; data curation, M.B.; writing—original draft preparation, M.U.; writing—review and editing, M.U.; visualization, M.U.; supervision, M.B. All authors have read and agreed to the published version of the manuscript.

**Funding:** This research received no external funding.

**Conflicts of Interest:** The authors declare no conflict of interest.

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
