# Peer review of "Assessment of Static Positioning Accuracy Using Low-Cost Smartphone GPS Devices for Geodetic Survey Points’ Determination and Monitoring"

_applsci, doi:10.3390/app10155308_

Round 1

Reviewer 1 Report

Dear Authors

I recognize that you have focused your research on a currently hot topic,

how to achieve precise postioning with cheap smartphone GNSS devices.

On the other hand your carried our experiments were not precisely described 

in your manuscript. Furthermore I detected some technical issues

and last but not least an uncountable number of grammar problems. I have divided my comments in a technical A) and a grammar part B). As you will see the grammar table is not complete and I recommended definitely to revise the manuscript with the help of an englich native speaker.

A) Technical Part

1) Title

   I propose to reconsider the title of your paper; you mention GNSS data,

but finally you show results based just on GPS data; Galileo could not be used, due to lack of reference data and in terms of GLONASS you just mention 'does not change results' without showing clear numbers.

Moreover you just show results of single baseline processing and not of

geodetic network monitoring;

Thus a more appropriate title might be:

'Assessment of positioning accuracy using low-cost smartphone GPS devices'

2) In the abstract the baseline length of 4 kilometers to the reference station

OPNT is mentioned, but I did not see this number later in the text; should also show up in the relevant chapters

3) References: later in the text you show some arguments why you base your calculations on the fast static model and not on RTK with VRS concept.

Therefore I wonder why the basic thesis text of Urs Wild (from AIUB Berne) who publishes first the FARA algorithm in the begin of the 1990s does not show up in the references section? Or does your 'fast static' approach not follow the FARA algorithm?

4) Introduction page 2 , line 69

I wonder why you mention that observations on a second frequency help to correct for tropospheric refraction? The other mentioned effects are Ok, but the troposphere is non-dispersive for L-Band.

5) Below figure 1 you note that only 14 GPs satellites (12 of block IIF) and 2 of block IIIF broadcast L5 signals, I assume you mean IIIA instead of IIIF, as the IIIF satellites will not be launched before 2025.

6) A major problem with your manuscript relates to the formulas (1)-(4)

and (5)+(6). The first group is correctly described as DD observations. In (5) and (6) you refer to raw measurements (ZD). But later on (see Figures 25) you show time series of Single differnce observations and point back to

eqautions (5) and (6). You have to definitely better describe at which stage of your processing you are working with DD,SD or ZD observations; its

quite clear that data assessment is based on ZD or SD data and coordinate processing on data at SD and DD level-but this has to be clearly stated.

7) concerning eq(5) and (6)

a) I missed a statement that this so-called geometry free LC is not the

geometry LC usually published in literature. Your LC is not a differences just of carriers or a difference just of codes- you work with a difference of phase-Code ; clarify to the readers

b) I do not see a comment how you handle the ionospheric delay in your

  N1 and N5 Linear Combination (or just difference) as the ionosphere is

 clearly different for the Code on the first and the 5th frequency. How

does this influence (bias) the time series in Figures 25 ?

8) at the end of page 4 you note

'researchers found that the number of VRS stations .. does not increase

accuracy..'

why do you mention the 'number'. The positioning quality is based on the

quality of the error models and due to distance correlation, how close the

VRS is located. The first effect f course introduces a potential bias but

the second point is always fulfilled-as for static RTK the VRS station is always

close.

9) Fast static versus RTK(VRS)

I can follow your notes at the end of chapter 2 but you should consider

that the bias introduced by poor error modelling with VRS is always

small at the 1-3cm level (which is not much compared to the problem

of poor smartphone carrier phase data processing). Moreover with VRS you can bridge a few tens of kilometers baseline to the next real reference stations. With fast statice you are limited to a few kilometers distance due to the influence of the ionosphere. Discuss the pors and cons

10) at the end of page 5 you mention that 'delivering raw dual-frequency data..' hepls for a more sophisticated processing at at DD level

You may process also single frequency data in DD mode; the dual frequency

just helps to reduce the ionospheric effect

11) In chapter 3 (line 222) you mention

' .. methods like ionospheric scintillations...'

Ionospheric scintillations are NO METHOD, scintillations are a physical effect

12) in chapter 4 you mention that its necessary to know the exact PC-Offsets;

you described the problem but unfortunately your paper does not cover any numbers or tables which shows the reader the PCO you have determined for

your smartphone antenna? Is this small effect really essential, gicen the

still poor smartphone phase data? if yes - show it !

13) your experiments and processing is described in chapter 5;

For me this chapter is too lengthy and shall be shortened just to contain

the main findings. The major problem for the reader is to identify if your tables show coordinate differences just form a processing of a 50cm baseline

or from 4km baselines between OPNT and the smartphone on the one hand

and between OPNT and the high quality receiver on the other.

Negative valuues in tables a mentioned as 'min' and positive as 'max'. Cosider if this is useful . It might much more useful to describe the

reason for coodinate matches in the cm range and also diff vectors of 0.5m.

14) In the begin of chapter 5 you note

'.. on a tripod and acted as a standalone device'. What do you mean

as a 'standalone device? I assume the experiments still calculates baselines

to OPNT and not single Point positions in 'standalone mode'?

15) which differences do you expect between mounting on a tripod with respect to the aluminium bean? or is the second mounting just for an easy

comparison to the high quality receiver data?

16) Figures 16/17 and 23/24 show skyplots for 2 receivers located 50cm

apart ? this makes absolutely no sense . Its obvious that the plots have to

be identical. why do you show both of these plots? to compare S/N ratio?

17) In chapter 6 you try to blame the reference station provider for not offering Galileo data. But its up to you to take care. If you feel Galileo data is necessary for your experiments-then you have to rerun the experiments before writing the paper; if not, then you should not blame the provider.

Btw. most reference stations in Europe offer currently 4 systems GNSS data.

18) Do you have any reasonable explanation why the time series of N1(P1) in Figure 25b (smart phone is noisy as expected, but not the the N1(P5)

series?

B) Grammar

  • Abstract , 1st sentence

Recent developments enable to access raw GNSS measurements of mobile phones.

  • Next sentence

Initially, researchers using signals gathered by mobile phones for high accuracy surveying were not successful in ambiguity fixing. Nowadays, GNSS chips…

  • 3 lines later

.. with a dual-frequency GNSS receiver…‘

  • Next sentence

For two sessions the mobile phone…- horizontally. At the same time a high-class geodetic..

  • 4 lines later

… sub-sessions to check the accuracy oft he surveying results in fast static mode.

  • 2 sentences later

‚In comparison to the fixed reference point position

The rest oft he sentence hast o re-phrased to become understandable

  • Introduction , line 7

‚Since then, the scientific community…‘

  • 3 lines later

‚.. running on an Android operating system.‘

  • Next paragraph line 1
  • ‚.. obtained by a smartphone device is about a few meters
  • Rephrase the text from

‚ To obtain precise positioning  - until – ‚and postprocessing positioning techniques.‘

  • Next pargarph line 1

‚High positioning accuracies using GNSS technology can be achieved with…‘

  • 9 lines later

‚The next significant technological breakthrough in smartphone positioning was in year 2017, when the Broadcom company released the first dual-frequency GNSS chipset…‘

  • 3 lines later

‚.. equipped with a dual-frequency GNSS chipset was the Xiaomi Mi 8 … dual frequency chipset. The ability to use …‘

  • 5 lines later

‚.. introducing RTK (Real-Time-Kinematic) or PPP (Precise Point Positioning) algorithms…

  • Next paragraph, line 3

‚Such a problem occured with GNSS chipsets built in mobile phones, such…‘

  • 2 lines later
  • Re-phrase sentence ‚This action caused…‘ ist not understandable
  • 9 lines later

‚.. This reserach work demonstrated that it is not possible to fix phase ambiguities and therefore to reach cm-accuracy in real-time can hardly be reached due tot he …

  • End of this paragraph

‚.. of current mobile phones is quite superior with respect to previous models..‘

  • Next paragraph, first sentence

‚The authors of this paper focused on the assessment of carrier…‘

  • Last sentence in Introduction

‚.. data we analyzed the signals, the quality…‘

  • Chapter 2 ,line 3
  • Here you note

‚Therefore the researchers have been …‘

Its quite unclear tot he reader who is meant with ‚the researchers‘ here?

Is it you – the authors? Then write ‚the authors‘ – or is it any other researcher?

  • Text below figure 1

‚As we can see from Figure1, the smartphone recorded the following GNSS data:

  • GPS L1/L5 carrier phase measurements transmitted at….
  • GPS pseudoranges P(L1) and P(L5)
  • Galileo E1/E5a carrier phase measurements
  • Galileo pseudoranges P(E1) and P(E5a)
  • GLONASS L1 carrier phase measurements and pseudoranges
  • Text below:

It should be noted that the GPS L1 and L5 frequencies correspond with Galileo E1 and e5A center frequencies

  • 3 lines later

‚.. write the following double-difference (DD) observation equations.‘

  • Text just above equations (5),(6)

‚In some papers mainly the signal to noise ratio has been analyzed, but we decided…‘

  • 10 lines below equ(6)

Please re-phrase the sentence

‚Based on fixing ambiguities on L1…‘

  • Next sentence

‚.. not pointed out by the manufacturers.‘

  • 10 lines later

‚The latest post-processing algorithms involve highly advanced statistical methods for integer

ambiguity determination.‘

  • 2 lines later

‚.. allow us for modelling the distance dependent errors…‘

  • 5 lines later
  • ‚.. virtual reference station data in any …‘

So as mentioned above I stop with my grammar commenst at the end of chapter 2, but there defintely more to correct

best regards

Reviewer 2 Report

The data set author collected is quite poor, only 3 hours of data. It would have been better to test for a longer period, e.g. 24 hours or more, possibly in different atmospheric conditions. So, author's conclusions should be more cautious.

Figure 1: did the smartphone also collect BEIDOU data? If not, author should explain why

190 Paragraph 3 title: I would use the term "Utilization" or "Use" rather than "Usefulness" and would omit "smartphone" because the paragraph presents the application of GNSS carrier phase observation acquired by any receiver, non necessarily a smartphone.

239 "utilization" rather than "usefulness"

252 a fixed control point

Author should write the distance of the station OPNT from the test site: how long is the test baseline? This is a very important parameter I can't find in the paper.

254 "performed" rather than "prepared"

Topcon Tools final version is 8.2.3, why did author use an older version? Magnet Tools software (the new updated version of Topcon Tools) would be more advisable, insted of converting an older RINEX format.

In the conclusion author should better write of a "some centimeters acuracy", not "centimeter" which means 1 cm or less. Also considering the short time span of the test data, author should be more cautious.

389 This could have a direct effect

Interesting experiment anyway, and the first part reporting the stare of the art is well written.

Round 2

Reviewer 1 Report

Dear Authors

your manuscript has been improved significantly during the first review cycle.

Please find below some further grammar hints and one still standing technical issue which you should address in a final version

A) Technical

the technical question relates to my topic 7 raised in the previous review cycle.

its related to your 'geometry free' LCs N1(P1) and N1(P5). I agree with your explanations, but these explanations did not focus on the main point in my comment. The main point is the handling of the ionospheric delay. You denote N1(P1) and N1(P5) as 'geometry free'. But geometry free means usually that the remaining part is the ionospheric delay. In your case you are not really interested in the ionosphere, you want to check the quality/noise of your P1 and P5 code observations. By the way your 'geometry free LC' covers the ionospheric effect twice as you built a combination of phase and code. As the ionospheric delay acts in both observations with different sign, the effect is doubled in your N1 variables. The reason why you need not to deal with the ionospheric delay is, that your observations are on the Double Difference level and therefore the ionospheric delay almost vanishes over 4km.

You should explain these circumstance to the readers.

B) some still  standing grammar hints

1) Abstract, line 17

'.. sessions at a known point location.'

2) line 24

'.. comparison to the fixed reference ...'

3) Introduction, line 35

'... When the first Android...'

4) line 78

  'This action limited the use of smartphone observations for precise

  positioning techniques such as RTK and PPP. Fortunately, Google...'

5) chapter 2 line 108

  'Therefore, recently scientists are paying special ...'

6) sentence below equation (6)

  '.. epoch N1(P1) and N1(P5) GPS-only combinations for geodetic...'

7) line 157

  '.. on obtaining a reliable and accurate...'

8) chapter 3 , line 191

  '.. topographic surveys, the static method...'

9) 5 lines later

  '.. typically require an accuracy between ...'

10) chapter 4, line 252

'.. research area with a GNSS Javad Alpha...'

11) line 255

'The first surveying test to assess the usefulness off ... on 15th of January 2020. Data was collected over 60 minutes and the mobile phone...'

12) line 268

 '.. purpose, where the mobile phone...'

13) line 272

  'smartphone was fixed in three...'

14) line 281

'... interrupted or of poor quality.'

15) line 284

  '.. for all three cases were obtained with millimeter level of precision. After calculating (please re-phase this last sentence properly !!!)

16) line 298

 'In this case the smartphone...'

17) 2 lines later

'.. experiment, the Javad receiver...'

18) chapter 5, line 305

  '.. and compared to the reference position.'

19) 4 lines later

  '.. 1-h session, the number of .... below 6, with an average number of 8 satellites.'

20) below figure 10, line 313

'The complete surveying session lasted...'

21) 3 lines later

'. and compared to the reference position.'

22) text below figure 15

  '.. we prepared a skyplot covering the GPS-satellites tracked by the

Huawei P30 Pro (Figure 16).'

23) line 328

  '.. 23rd of January, the Huawei... '

24) 2 lines later

 'Similarly, we mounted the Javad Alpha geodetic receiver at the the same place.'

25) line 336

 I do not understand the sentence 'Only for the first 15 and 20 ...???

in the previous sentence you write that also for the 15min and 20min sessions all ambiguties were just float (not fixed). And now you write that

they were fixed ??? clarify

26) chapter 6 discussion, line 361

'.. of residual values of float DD geometric free...'

27) line 376

please change 'the line' to a more clear term like 'the x-axis'

28) line 383

  '.. with observation periods of just a few minutes are therefore..'

best regards
